# Evaluation of the Complex p.[Leu467Phe;Phe508del] CFTR Allele in the Intestinal Organoids Model: Implications for Therapy

**DOI:** 10.3390/ijms231810377

**Published:** 2022-09-08

**Authors:** Elena Kondratyeva, Anna Efremova, Yuliya Melyanovskaya, Anna Voronkova, Alexander Polyakov, Nataliya Bulatenko, Tagui Adyan, Viktoriya Sherman, Valeriia Kovalskaia, Nika Petrova, Marina Starinova, Tatiana Bukharova, Sergei Kutsev, Dmitry Goldshtein

**Affiliations:** Research Centre for Medical Genetics, 115522 Moscow, Russia

**Keywords:** cystic fibrosis (CF), CFTR, complex allele [L467F;F508del], intestinal current measurements (ICM), intestinal organoids, CFTR modulators

## Abstract

In the cohort of Russian patients with cystic fibrosis, the p.[Leu467Phe;Phe508del] complex allele (legacy name [L467F;F508del]) of the CFTR gene is understudied. In this research, we present the results of frequency evaluation of the [L467F;F508del] complex allele in the Russian Federation among patients with a F508del/F508del genotype, its effect on the clinical course of cystic fibrosis, the intestinal epithelium ionic channel function, and the effectiveness of target therapy. The frequency of the [L467F;F508del] complex allele among patients with homozygous F508del was determined with multiplex ligase-dependent probe amplification followed by polymerase chain reaction and fragment analysis. The function of ionic channels, including the residual CFTR function, and the effectiveness of CFTR modulators was analyzed using intestinal current measurements on rectal biopsy samples and the forskolin-induced swelling assay on organoids. The results showed that the F508del/[L467F;F508del] genotype is present in 8.2% of all Russian patients with F508del in a homozygous state. The clinical course of the disease in patients with the F508del/[L467F;F508del] genotype is severe and does not vary from the course in the cohort with homozygous F508del, although the CFTR channel function is significantly lower. For patients with the F508del/[L467F;F508del] genotype, we can recommend targeted therapy using a combined ivacaftor + tezacaftor + elexacaftor medication.

## 1. Introduction

Cystic fibrosis (CF) is an autosomal recessive disorder caused by mutations in the cystic fibrosis transmembrane conductance regulator (CFTR) gene. The product of the gene is an ionic channel carrying chloride ions through the membranes of epithelial cells. Mutations in the CFTR gene affecting the function of chloride channels lead to impairment of fluid transport regulation in the epithelial tissue of the lungs, pancreas, and other organs, which leads to polyorganic failure. The significant variety of CF symptoms in patients can be caused by multiple factors, including the variety of CFTR genotypes, the effect of modifier genes, and external factors [1].

As of date (29 April 2022), more than 2000 mutations or nucleotide sequence variants and approximately 250 polymorphisms in the CFTR gene were detected [https://www.cftr2.org/, accessed on 7 June 2022]. Pathogenicity was established only for 401 genetic variants, 49 variants with various clinical consequences, and 11 variants of unknown significance.

CF diagnostic process is complicated by a possible presence of complex alleles. An allele is called complex if it carries at least two mutations in the CFTR gene, in which case each pathogenic variant might affect separate stages of CFTR biogenesis. The frequency of complex alleles is currently undetermined. According to our research results, the frequency of the complex allele [L467F;F508del] (c.[1399C > T;1521_1523delCTT], p.[Leu467Phe;Phe508del]) in Russian patients with a homozygous F508del variant is 8–10% [2].

The introduction of novel therapeutic approaches, abating lung inflammation, cirrhosis, and other CF complications, substantially prolonged patients’ life expectancy. From a once fatal disorder, CF became chronic; however, median life expectancy in developed countries varies from 20 to 40 years and more [3,4]. The most notable result of CF pharmacotherapy development is the modern strategy of CFTR modulator target therapy aimed at restoration of the structure and function of the CFTR protein. The effectiveness of CFTR modulators is evaluated based on the ability to increase the amount of the CFTR protein on the surface of epithelial cells and/or enhance its function [5,6].

Two simultaneous mutations at the same allele (complex allele) may reduce the effectiveness of target therapy by increasing pathogenicity of each other. As an example, an additional cis-variant may increase CFTR proteolysis or increase folding disorders. Furthermore, the presence of an additional cis-variant may lead to the changes in the structure of protein. These changes in the protein’s structure, in return, may lead to decrease or inability of protein’s interaction with CFTR modulators, resulting in the reduced effectiveness of CTFR modulators [7].

Patients with the F508del/F508del genotype are problematic due to significant variability of responses to ivacaftor + lumacaftor therapy [8,9], which can be explained by the presence of complex alleles (additional cis variants). In this case, the cis variant, often not pathogenic on its own, may reduce the overall amount of the CFTR protein available for the corrector or alter the 3D structure of the protein molecule, affect the sites of interaction between the protein and the corrector, and nullify its effect [7]. For example, A. Masson et al. found the presence of the complex [F508del;F87L;I1027T] allele to correlate with the absence of response to ivacaftor + lumacaftor in the cohort of patients with F508del in a homozygous state, while in patients with positive responses to ivacaftor + lumacaftor no complex alleles were detected [9]. N. Baatallah et al. showed on the HEK293 cell line that the presence of the [L467F;F508del] complex allele correlates with nearly total loss of the CFTR protein, in which case the CFTR corrector VX-809 (lumacaftor) does not provide a positive effect [10].

Thus, for patients carrying complex alleles, effective target therapy prescription may be a complicated task. Although there is sufficient information on complex alleles of the CFTR gene in the literature, the question of the influence of complex alleles on the effectiveness of target therapy is understudied and their frequency in various populations of patients requires further examination.

Study aims: to examine the frequency of the p.[Leu467Phe;Phe508del] complex allele in the Russian Federation in patients with a F508del/F508del genotype, its effect on the clinical course of cystic fibrosis and the effectiveness of target therapy on a model of intestinal organoids.

## 2. Results

### 2.1. Clinical Picture 

The examination of the frequency of the [L467F;F508del] complex allele among patients with a homozygous F508del genetic variant in Russian Federation showed that 8.2% (65 out of 810 patients) have a c.1399C > T (p.Leu467Phe) nucleotide sequence variant on the same allele as a c.1521_1523del (p.Phe508del) variant. Moreover, in 0.49% (4 patients) the c.1399C > T variant was detected in a homozygous state, which indicates the presence of the [L467F;F508del] complex allele on both chromosomes.

The data analysis of patients from the register of year 2020 showed (Table 1) that both examined groups (with F508del/[L467F;508del] and 508del/F508del genotypes) did not differ in the age of diagnosis, respiratory function characteristics, and nutritive status and had an identical microbiome and frequency of exacerbations in 2019, received similar treatment, including courses of intravenous antibacterial therapy.

### 2.2. Evaluation of Apical Membrane Ionic Channel Functional Activity in Patients with the Complex [L467F;F508del] Allele Using Intestinal Current Measurements (ICM)

For patients with a CF diagnosis with a F508del/[L467F;F508del] genotype, we carried out ICM analysis (three patients, Table 2) in comparison with the homozygous F508del/F508del group (Table 2, Figure 1) and a control group (Table 2).

In the group of patients with a F508del/[L467F;F508del] genotype (Figure 1C), the short-circuit current density (ΔISC) as a response to amiloride (sodium channel stimulation) was −11.39 ± 0.79 µA/cm^2^. The ΔISC change as a response to forskolin (chloride channel stimulation) was 2.5 ± 0.61 µA/cm^2^. As a response to histamine, ΔISC changes to negative, which indicates the influx of potassium ions into the cells. With that, the current density was 12.39 ± 1.98 µA/cm^2^.

Comparing the data of patients to the figures in the control group, the homozygous F508del group, and the group of patients with two class I mutations in the genotype, we showed that in the group of patients with a complex [L467F;F508del] allele, the response to amiloride was higher than in the control group and the class I/class I group but lower than in the F508del/F508del group (Table 2). The ΔISC change as a response to forskolin was lower in patients with the [L467F;F508del] genetic variant than in the homozygous F508del/F508del group (Figure 1B) and the control group (Figure 1A) but higher than in the class I/class I group. With that, the response to histamine in patients with [L467F;F508del] in the genotype was also lower than in the F508del/F508del group and the control group but higher than in the class I/class I group.

Thus, using the ICM method, we showed that the chloride channel (CFTR) function is absent in case of a [L467F;F508del] complex allele, and the response to forskolin/IBMX is lower than in patients with a homozygous F508del genetic variant.

### 2.3. Evaluation of the Effectiveness of CFTR Modulators in Patients with the Complex F508del/[L467F;F508del] Allele on the Intestinal Organoids

The effect of CFTR modulators on the functional activity of the CFTR protein was studied in intestinal organoid cultures received from three patients with a F508del/[L467F;F508del] genotype.

The response to forskolin allows the determination of the residual function of the CFTR channel. In case of “severe” genotypes, including F508del/F508del, the organoids were treated with forskolin (Fsk) at a high concentration of 5 µM for 1 h. The residual functional activity of the CFTR channel in patients with the complex allele was insignificant (Figure 2 and Figure 3) and the AUC (area under the curve) was on average 253.43 ± 85.85. The obtained values were practically identical to the control 382.27 ± 106.36 (Figure 2). The VX-770 potentiator (ivacaftor) increases the functional activity of the CFTR channel in the control group to 818.1 ± 197.75 (5 µM Fsk, 1 h). For intestinal organoid cultures 1–3 with a complex allele, VX-770 proved to be approximately two times less effective under the same conditions, and the average AUC values were 504.72 ± 48.55. The VX-809 corrector (lumacaftor) increased the functional CFTR activity from 382.27 ± 106.36 to 1119.52 ± 116.41 r.u. in the control group. In the examined group with the complex allele, its effects were approximately 2–3 times weaker and the obtained AUC values were, on average, 533.18 ± 143.51 (Figure 2).

In the control culture, as expected, VX-770 + VX-809 and VX-770 + VX-661 (ivacaftor + tezacaftor) effectively increased the amount of functional CFTR on membranes of epithelial cells (Figure 3) and the AUC values were, respectively, 2165.93 ± 67.64 and 1773.17 ± 113.69 (Figure 2). In case of cultures 1–3, which carry the F508del/[L467F;F508del] genotype, the effect of combined VX-770 + VX-809 and VX-770 + VX-661 treatment was on average 2 times lower (Figure 3): 1137.15 ± 403.54 and 898.44 ± 372.65, respectively (Figure 2). Aside from that, we detected a significant variability in the examined group; for example, the response to VX-770 + VX-661 for culture 1 the AUC was 479.65 ± 35.03, culture 2 it was 1022.23 ± 195.83, and for culture 3 it was 1193.44 ± 35.99. With the AUC values lower than 1000, the therapeutic effect of the prescribed medication is known to be insignificant or absent [11]. 

For the patient group with the complex [L467F;F508del] allele in the F508del/[L467F;F508del] genotype, we showed effective restoration of the CFTR function in case of simultaneous treatment with the three-component CFTR modulator VX-770 + VX-661 + VX-445 (ivacaftor + tezacaftor + elexacaftor). The AUC values after 1 h of 5 µM Fsk treatment for three cultures of intestinal organoids exceeded 1000 r.u., reaching 2757.12 ± 219.79, 4219.31 ± 168.22, and 3539.18 ± 74.51 (on average 3505.20 ± 731.68), and were generally comparable to the results obtained for the control F508del/F508del culture—4717.95 ± 112.03. Culture 2 and control showed similar values when treated with VX-770 + VX-661 + VX-445 (Figure 2 and Figure 3). 

It is worth noting that we observed a wide range of responses to CFTR modulators in the examined group with the complex allele. For example, culture 1 of intestinal organoids showed the weakest response out of all three cultures to the combined medications VX-770 + VX-809, VX-770 + VX-661, and VX-770 + VX-661 + VX-445 (Figure 2). 

When F508del/[L467F;F508del] organoids were treated by forskolin at 0.128 μM concentration the effects of VX-770, VX-809, VX-770 + VX-809, and VX-770 + VX-661 during the rescue of the CFTR function were insignificant and the AUC values did not exceed 1000 (Figure 2). Only the combination of VX-770 + VX-661 + VX-445 resulted in the rescue of CFTR function (Figure 2, organoid culture 2 and 3)

## 3. Discussion

Complex alleles were initially described by N. Kalin et al. in 1992 [12], in particular, a F508C-S1251N allele, which affects the CFTR function. Each pathogenic variant from the complex allele may affect separate stages of CFTR protein synthesis. Complex alleles may be classified as disease-causing, neutral, or as modifying therapy effectiveness [7]. 

The problem of complex alleles provides a challenge of determining each separate mutation’s pathogenicity, also calling into question the pathogenicity of already known mutations. The exact number of complex alleles is unknown and most likely to be much higher than described. In a large-scale study in 2016, B. Vecchio-Pagán et al. provided data on 21 complex allele detected in the USA in patients carrying the F508del genetic variant and determined the frequency of these alleles among patients with CF [13].

The L997F variant has been identified in two patients either as a separate variant on one of the alleles, or in a combination with R117L. The clinical picture of the patients showed that the L997F variant on its own (in combination with a pathogenic *CFTR* variant in the second parental allele) may cause CFTR-associated disorders, and the complex [L997F;R117L] allele is linked to CF with a mild phenotype [14]. The R117H and [TG12T5] (intronic splice site mutation) variants in a cis state enhance each other’s pathogenicity and lead to CF, while each of these variants separately in a combination with a pathogenic variant in the other parental allele causes CFTR-associated disorders [15].

The presence of an additional amino acid replacement, L467F, in a cis state with F508del (in comparison to only F508del) leads to a significant decrease in CFTR activity due to a serious maturation defect. The expressed immature glycosylated CFTR protein was not repaired by correctors, being insusceptible not only to the lumacaftor but also to the more effective combination of elexacaftor + tezacaftor [16].

Because of the high prevalence of complex alleles in populations, the accumulation of data on their frequency and effect on therapy is essential for understanding the algorithms of target medication prescription and for adding these alleles to DNA diagnostic panels, as well as for CFTR modulator therapy preparation to exclude unnecessary therapy costs and the toxic effect of the medication.

The current study showed the high frequency (8.2%) of the complex [L467F;508del] allele in a population of patients with a F508del/F508del genotype (72% of patients examined) in the Russian Federation in comparison with the data obtained for the North American patient population [13]. The results served as a basis for recommending analysis to identify the L467F variant in patients with homozygous F508del prior to prescribing therapy with combined medication—ivacaftor + lumacaftor (VX-770 + VX-809), ivacaftor + tezacaftor (VX-770 + VX-661).

When comparing the examined groups with homozygous F508del and compound heterozygous F508del and [L467F;508del], we did not detect a statistically significant difference in the course of the disease, exacerbations, or prescribed therapy. These results were obtained for the first time. 

The presence of [L467F;508del] probably does not affect the severity of the disease. It has been shown previously that non-pathogenic *CFTR* genetic variants in one complex allele may not affect the symptoms of the disorder (be neutral) or acquire pathogenic properties in a combination (form a pathogenic phenotype). For example, additive pathogenic effects were shown for various complex alleles, such as the triple [R74W;V201M;D1270N] allele, which results in a partial skip of exon 3 because of the R74W mutation and the formation of a protein with decreased functional activity because of the D1270N and V201M mutations [17]. The R74W and D1270N variants, combined with a pathogenic variant in the other parental allele (for example, F508del), may be insufficient to cause CF or a CFTR-associated disease. However, the complex [R74W;V201M;D1270N] allele is pathogenic and causes CFTR-associated diseases [17]. 

Using ICM method, we showed that the complex [L467F;508del] allele negatively affects the function of chloride and sodium channels in comparison to the values in patients from the group with homozygous F508del, while in patients with a class I/class I genotype the function of sodium and chloride channels is impaired to a greater extent. The presence of [L467F;508del] affects the structure and quantity of the protein and disrupts the interaction with CFTR modulators, lowering their effectiveness, but does not increase the severity of the disease’s symptoms. 

A. Masson et al. [9] showed that patients with a complex [F87L;F508del;I1027T] allele did not respond to therapy with ivacaftor + lumacaftor (VX-770 + VX-809). Another study provides data of two CF patients with compound heterozygous F508del and a minimal function variant who did not show any positive effects after treatment with the triple elexacaftor/tezacaftor/ivacaftor combination [16]. The functional analysis of nasal epithelium obtained from these patients confirmed the absence of response to pharmacological treatment. A molecular genetic examination detected the presence of an additional amino acid replacement L467F in a cis state with the F508del variant, and functional and biochemical analysis showed that the effect of CFTR modulators on patients with the complex [L467F;508del] allele is very low.

Forskolin-induced swelling (FIS) assay on intestinal organoids acquired from CF patients with different CFTR genotypes, including complex alleles, allows the specific examination of the residual function of the CFTR channel, as well as determine the effect of CFTR modulators (target drugs) on the CFTR channel function restoration. The results of the FIS assay may indicate the effectiveness of a highly effective modulator therapy and could provide guidance in prescribing target therapy to patients [18]. The FIS assay is based on the activation of the CFTR channel by forskolin, which increases the concentration of intracellular cAMP. In intestinal organoids, the spherical structures of one-layer epithelium, the activation of the CFTR channel leads to swelling due to the transport of chloride ions into the internal cavity of the organoids (lumen). The rate and extent of organoid swelling represents the functional activity of the CFTR protein in patients with CF [19].

There is no data on the examination of complex alleles on a model of intestinal organoids in the literature. Nevertheless, J. Dekkers et al., in 2016 [20], showed on a culture of intestinal organoids obtained from patients with a F508del/F508del genotype that there was a significant variability of response (restoration of CFTR function) in the case of treatment with combined ivacaftor and lumacaftor, which may be explained by the presence of complex alleles. It was shown on an intestinal organoids model that the VX-770 potentiator and the VX-809 corrector separately and insignificantly increased the functional activity of the CFTR channel in case of F508del/F508del genotype. The combination of these CFTR modulators was shown to restore the CFTR channel functional activity effectively; therefore, ivacaftor + lumacaftor (VX-770 + VX-809) are prescribed to patients with the F508del/F508del genotype [8,20] (https://www.cff.org/managing-cf/cftr-modulator-therapies accessed on 10 June 2022).

In the current study, it was first shown on an intestinal organoids that the combined VX-770 + VX-809 (ivacaftor + lumacaftor) is approximately two times less effective in restoring the functional CFTR activity for the F508del/[L467F;F508del] genotype compared to F508del/F508del; therefore, the ivacaftor + lumacaftor therapy is not recommended for patients with a F508del/[L467F;F508del] genotype. The effect of VX-770 + VX-661 (ivacaftor + tezacaftor) on the CFTR function was even less pronounced in comparison to VX-770 + VX-809. It is known that if the AUC values of FIS do not exceed 1000 (results of the FIS assay) after treatment with target drugs, the latter do not have a significant therapeutic effect on the patient [13].

N. Baatallah et al. showed [10] on a HEK293 cell line transfected with plasmids with a normal (WT) genetic variant of the *CFTR* gene, as well as mutant L467F and F508del variants, using Western blotting, that the proportion of functional protein (mature, fully glycosylated) for the mentioned variants is 86% (WT), ~40% (L467F), and 22% (F508del). Thus, in case of the F508del/F508del genotype, both correctors and potentiators have a target for action. However, in the complex [L467F;F508del] allele, the additional L467F mutation, which on its own lowers the amount of mature protein approximately two times compared to WT, inhibits the functional CFTR protein formation almost completely. In this case, CFTR modulators (shown for VX-809 [10]) do not have a target for action, and the CFTR potentiator and corrector therapy cannot be effective.

The presence of at least one F508del genetic variant in the patient’s genotype necessitates the prescription of the three-component ivacaftor + tezacaftor + elexacaftor (VX-770 + VX-661 + VX-445) medication (https://www.cff.org/managing-cf/cftr-modulator-therapies accessed on 10 June 2022). It is worth noting that using VX-770 + VX-661 + VX-445 on a model of intestinal organoids showed very high the AUC values (>3000) for each examined culture. It is known that AUC values higher than 1000 indicate a high probability of a therapeutic effect of a target drug prescribed to the patient. Based on the obtained results, we concluded that the positive effect of the triple VX-770 + VX-661 + VX-445 therapy on the restoration of functional CFTR in intestinal organoid cultures with a F508del/[L467F;F508del] genotype will be effective in everyday clinical practice. At the moment, the research continues.

## 4. Materials and Methods

### 4.1. Estimation of the Frequency of the Complex Allele [L467F;F508del] 

To establish the frequency of the complex [L467F;F508del] allele in the Russian Federation, we analyzed DNA samples of patients aged from 2 to 39 years with a “cystic fibrosis caused by a F508del mutation in a homozygous state” diagnosis—810 patients in total, which comprises 21.3% of all patients from the Russian CF register and 72.5% of patients with a F508del/F508del genotype from the Russian CF register in 2020 [21].

All samples were obtained via DNA extraction from whole venous blood; the biological material was taken in CF centers in various regions of the Russian Federation.

The patients’ DNA was analyzed using multiplex ligase-dependent probe amplification (MLPA) followed by polymerase chain reaction (PCR) and fragment analysis.

### 4.2. Criteria for Inclusion of Patients in the Study. Description of the Clinical Picture

The criteria for inclusion of patients with CF were the diagnosis of “cystic fibrosis” confirmed according to the national consensus [1], the consent of the patient and/or parents to comply with the terms of the study, and signed informed consent. The criteria for inclusion of healthy people were the absence of cystic fibrosis, the consent of the patient to comply with the terms of the study, and signed informed consent.

The study and the informed consent form were approved by the Ethics committee of the Research Centre for Medical Genetics, Ministry of Science and Education, Russian Federation on 15 October 2018 (Ethics committee chairman—Prof. L. F. Kurilo).

To evaluate the condition and describe the patients’ clinical picture, we used data from the 2020 register of patients with cystic fibrosis [21]; the information was collected from medical history and ambulatory records of patients from various CF centers in the Russian Federation. The register format corresponded to the European CF patient register [22]. We carried out comparative data analysis of Russian CF patients with homozygous F508del in two groups: I—with a complex [L467F;F508del] allele (50 patients), and II—without a complex allele (479 patients). In both groups, the male to female ratio was 1:1. Both groups were comparable by examination age: in group I—8.9 ± 4.9 years, in group II—8.3 ± 4.6 years.

We evaluated the following clinical characteristics: age at the time of examination, age of diagnosis, proportion of patients diagnosed as a result of neonatal screening, proportion of patients with meconium ileus in their clinical history, microbiome, external respiration function indicators, nutritive status, and CF exacerbations in 2019 (CF-related diabetes (CFRD), allergic bronchopulmonary aspergillosis (ABPA), distal intestinal obstruction syndrome (DIOS), salt wasting syndrome (SWS), pneumothorax, liver disease, upper respiratory tract polyposis, pulmonary hemorrhage, osteoporosis, amyloidosis, and malignant tumors), as well as the amount of therapy, including the duration of intravenous antibacterial therapy.

The presence of chronic *Pseudomonas aeruginosa* infection or other Gram-negative bacteria was determined according to the modified Leeds criteria [23].

The criteria used for ABPA diagnostics were acute or subacute clinical manifestations (coughing, dyspnea, exercise intolerance, exercise-induced asthma, decrease in pulmonary function values, or increase in produced sputum amounts) not linked to other causes; general IgE > 500 IU/mL; positive skin prick test for aspergillus antigen (>3 mm) or positive specific IgE to *A.fumigatus*; precipitins to *A.fumigatus* or in vitro confirmed IgG antibodies to *A.fumigatus*; or novel (fresh) changes on pulmonary radiograph (infiltrates or mucus plugs) or characteristic changes on chest CT, which do not disappear after antibacterial therapy and standard kinesiotherapy [24].

To detect liver disease, we used criteria from the Great Britain register. These criteria allow to differentiate patients with severe liver disease (with portal hypertension) from patients with moderate liver disease (cirrhosis without portal hypertension). Liver disease without cirrhosis includes steatosis or viral hepatitis (https://www.cysticfibrosis.org.uk/the-work-we-do/uk-cf-registry accessed on 23 May 2021).

To describe the patients’ health characteristics, we used the following methods:

The CF patients’ nutritive status was evaluated using body mass index (BMI) by Quetelet (mass (kg)/height (m)^2^). The nutritive status (BMI) of the children was evaluated using the percentile system [25,26,27].

### 4.3. CFTR Function Analyses

To study the function of the CFTR channel via ICM, we included three CF patients with a F508del/[L467F;F508del] genotype (11:10:17 years; m:f:m) and three patients with homozygous F508del (17:17:15 years; f:m:f) for comparison. The control group for the ICM method consisted of 18 healthy people aged from 4 to 62 years. 

Using cultures of intestinal organoids, we analyzed the effect of the ivacaftor potentiator (VX-770), lumacaftor corrector (VX-809), and combinations of ivacaftor + lumacaftor (VX-770 + VX-809), ivacaftor + tezacaftor (VX-770 + VX-661), and ivacaftor + tezacaftor + elexacaftor (VX-770 + VX-661 + VX-445). The culture of intestinal organoids obtained from the female patient born in 2006 with a F508del/F508del genotype was used as control in the current study. The absence of additional mutations in the *CFTR* gene in the control patient with a F508del/F508del genotype for the intestinal organoid method was confirmed by Sanger sequencing.

The respiratory function was analyzed using the data on forced vital capacity (FVC) and forced expiratory volume during 1 s (FEV_1_) in a group of children able to perform the respiratory maneuver during spirometry [28,29].

For CF diagnostics and CFTR modulator therapy effectiveness evaluation, we used the method of intestinal current measurement (ICM) [30,31,32].

Colon biopsy was carried out on Olympus Disposable EndoTherapy EndoJaw Biopsy forceps (model #FB-23OU). The ICM examination was carried out according to the standard European operational procedures V2.7_26.10.11 (SOP) [33,34], following the algorithm described below.

The first stage was calibrating each recirculation chamber separately on the VCC MC 8B421 Physiologic Instrument, San Diego, USA. Problems, such as air in agar contact tips and fluid resistance, were eliminated during chamber preparation. Noise sources: vibrations near the equipment, accidental contacts with electrodes, and foreign working devices in the office were eliminated in so far as possible. The second stage, after calibrating the equipment, consisted of placing the rectal biopsy material into the chamber. The sample size was 3–5 mm. The material was placed into a special slider, which was then inserted into the chamber. The chambers were filled with a Meyler buffer solution. All reagents were obtained from Sigma-Aldrich, Merck. The pre-prepared buffer included: 105 mM NaCl, 4.7 mM KCl, 1.3 mM CaCl_2_.6H_2_O, 20.2 mM NaHCO_3_, 0.4 mM NaH_2_PO_4_.H_2_O, 0.3 mM Na_2_HPO_4_, 1.0 mM MgCl_2_.6H_2_O, 10 mM HEPES, and 10 mM d-glucose, as well as 0.01 mM of indomethacin. The analysis registration started by recording the basal short-circuit current (µA/cm^2^) (pre-amiloride stage). 

During the third stage, we added reagents in the following sequence: amiloride (an inhibitor amiloride-sensitive electrogenic sodium absorption), forskolin/IBMX (to activate cAMP-dependent CFTR chloride secretion), genistein (chloride channel), carbachol (to stimulate cholinergic calcium- and protein kinase C-mediated CFTR chloride secretion), 4,4′-diisothiocyano-2,2′-stilbene-disulfonic acid, DIDS (to inhibit DIDS-sensitive non-CFTR chloride channels), and ultimately, histamine (to determine the DIDS-insensitive component of calcium dependent chloride secretion). The analysis was terminated after recording the final resistance of tissue (Rt, ohm.cm^2^). The control group consisted of healthy volunteers. The CF patients with homozygous F508del comprised the comparison group (F508del/F508del) [30], the class I/class I group consisted of patients with class 1 variants in the genotype: 2143delT/712-1G- > T, G542X/R785X, c.264_268del/ 3271 + 1G > T.

The methods of obtaining stable cultures of intestinal organoids and the FIS assay were previously described in detail in studies by E. Kondratyeva (2020) and E. Kondratyeva (2021) [35,36]. The methods was based on protocols developed by J. M. Beekman’s guidance [20,37,38]. The intestinal organoids were obtained from samples of rectal biopsy, cultivated in “Matrigel” (Corning, Corning, NY, USA), passed once every 5–7 days. For the forskolin-induced swelling assay, the organoids were seeded onto 96-well plates and correctors VX-809, VX-661, and VX-445 (3.5 µM; Selleckchem, Houston, TX, USA) were added. After 24 h, the organoids were stained with Calcein (Biotium). The VX-770 potentiator in a concentration of 3.5 µM (Selleckchem, Houston, TX, USA) was added simultaneously with forskolin (0.128 or 5 µM). The intestinal organoids were incubated with forskolin and CFTR modulators for 1 h, and simultaneously at an interval of 10 min the pre-chosen fields were captured using a fluorescent Axio Observer 7 microscope (Zeiss, Jena, Germany). The organoids swelling was evaluated using the Image J program, and the obtained data was processed with Sigma Plot 12.5.

### 4.4. Statistics

The statistical processing of the data was carried out using an IBM SPSS Statistics 26 software package. Depending on the distribution, the measures of central tendency and dispersion were mean value (M) ± standard deviation (SD) or median (Me) (interquartile range)/(Q1;Q3). The statistical processing was carried out using the Mann–Whitney U criterion, Chi-squared test, and Fisher’s exact test. The differences were considered statistically significant in cases of *p* < 0.05.

## 5. Conclusions

The examination results showed that the F508del/[L467F;F508del] genotype was present in 8.2% of Russian patients with the F508del mutation in a homozygous state. The clinical course of the disease in patients with the F508del/[L467F;F508del] genotype corresponds to “severe” and does not vary from the homozygous F508del group, while the chloride channel function was lower when measured using ICM method. On the example of this complex allele, we showed that prescribing target therapy to CF patients with complex alleles can be a challenging task. For patients with the F508del/[L467F;F508del] genotype, we can recommend pathogenetic therapy with combined CFTR modulator ivacaftor + tezacaftor + elexacaftor (VX-770 + VX-661 + VX-445). Prescribing combined CFTR modulators, such as ivacaftor + lumacaftor (VX-770 + VX-809) and ivacaftor + tezacaftor (VX-770 + VX-661), is not recommended for patients with the F508del/[L467F;F508del] genotype because the absence of therapeutic effects is highly probable. The high prevalence of complex alleles, which lower the effectiveness of therapy with CFTR modulators, shows the necessity of their detection prior to prescribing target therapy in case of CF.

## Figures and Tables

**Figure 1 ijms-23-10377-f001:**
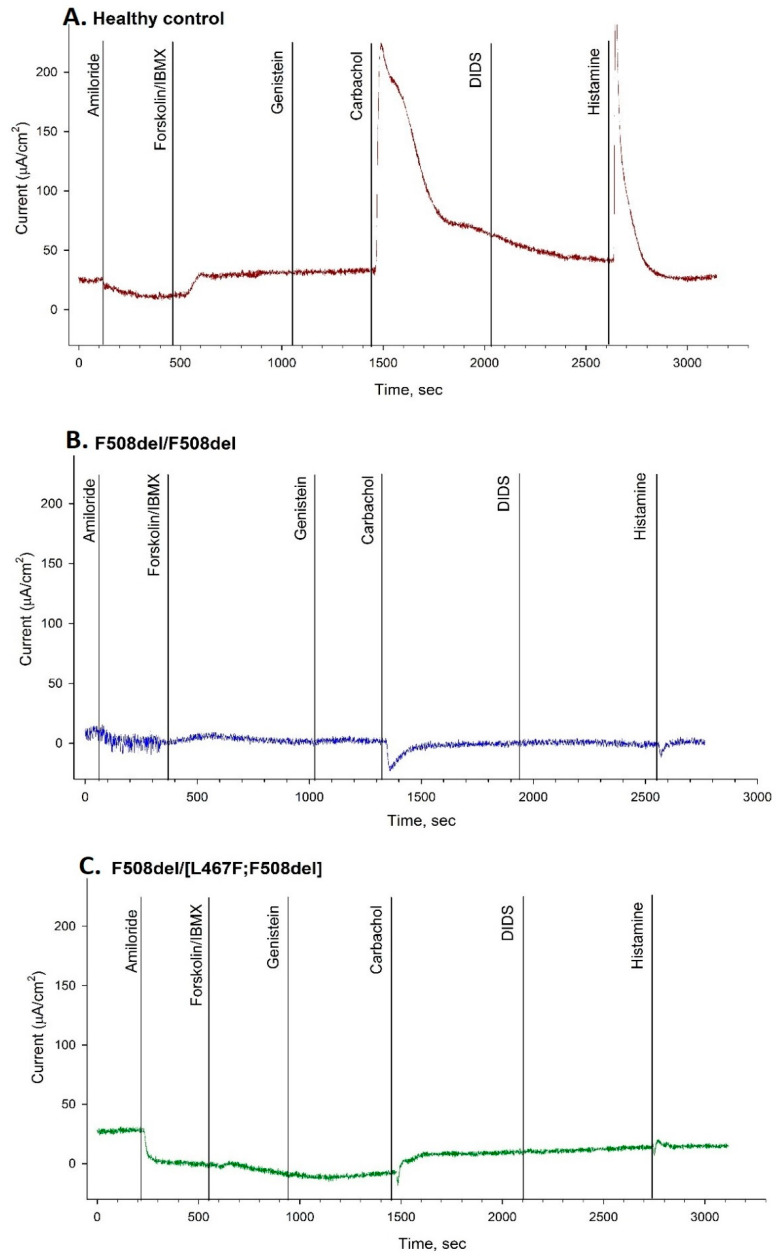
Intestinal current measurement (ICM); (**A**) Healthy control. The addition of amiloride caused a decrease in the short-circuit current density (ΔISC) was detected, as a response to forskolin/IBMX ΔISC increased significantly, and as a response to histamine ΔISC changed to positive; (**B**) Patient with a F508del/F508del genotype. The addition of amiloride caused a decrease in ΔISC was detected, no response to forskolin/IBMX, and as a response to histamine ΔISC changed to negative; (**C**) Patient with a F508del/[L467F;F508del] genotype. The addition of amiloride caused a decrease in the ΔISC, no response to forskolin/IBMX, and as a response to carbachol and histamine, ΔISC changed to negative.

**Figure 2 ijms-23-10377-f002:**
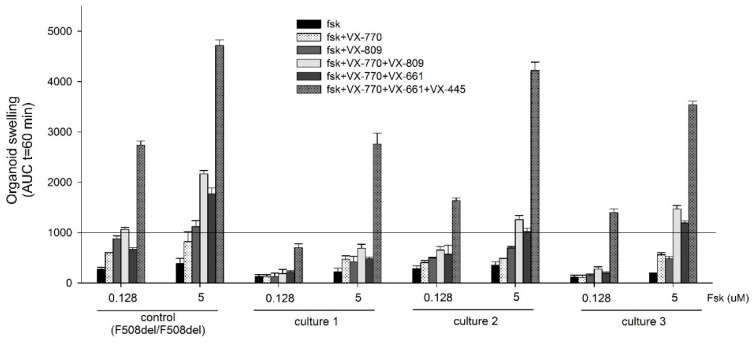
FIS assay results for cultures 1–3 of intestinal organoids with a F508del/[L467F;F508del] genotype compared to F508del/F508del (control).

**Figure 3 ijms-23-10377-f003:**
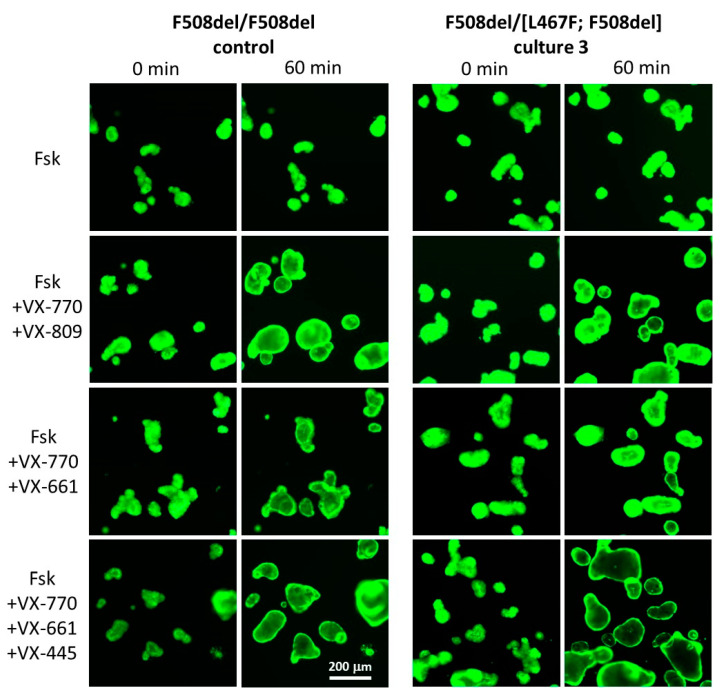
Characteristic images of intestinal organoids (on the example of culture 3) with a complex [L467F;F508del] allele before and after treatment with forskolin (5 µM) and CFTR modulators in comparison to the F508del/F508del control. Staining—Calcein (0.84 µM, 1 h), objective lens ×5, scale—200 µm.

**Table 1 ijms-23-10377-t001:** Comparative characteristic of patients with F508del/[L467F;508del] and 508del/F508del genotypes.

Characteristic	*n*	Group I F508del/[L467;F508del]	*n*	Group II F508del/F508del	*p*
Age, years					
M ± SD	50	8.9 ± 4.9	479	8.3 ± 4.6	*p* = 0.308
Me (IQR); Me (Q25;Q75)	50	8.4 (8.9); 8.4 (4.5; 13.2)	479	8.4 (7.1); 8.4 (4.5; 11.6)	
Sex					
Male	25	50.0%	235	49.1%	*p* = 0.899
Female	25	50.0%	244	50.9%	
Age of diagnosis, years					
Me (IQR); Me (Q25;Q75)	50	0.2 (0.6); 0.2 (0.1; 0.7)	475	0.2 (0.4); 0.2 (0.1; 0.5)	*p* = 0.352
Diagnosis based on neonatal screening, %					
No data	13	26.0%	104	22.6%	*p* = 0.463
Yes, positive	37	74.0%	345	74.8%	
Yes, negative	0	0.0%	12	2.6%	
Meconium ileus, %					
Total	6	12.0%	64	13.5%	*p* = 0.493
Surgical intervention	5	10.0%	61	12.9%	
Conservative therapy	1	2.0%	3	0.6%	
Microbiological analysis					
*Chronic P.aeruginosa*	13	27.1%	145	31.0%	*p* = 0.570
*Intermittent P.aeruginosa*	9	18.8%	99	21.4%	*p* = 0.666
*Chronic S.aureus*	26	55.3%	317	67.4%	*p* = 0.093
*Chronic B.cepacia*	2	4.3%	16	3.4%	*p* = 0.766
*Haemophilus influenzae*	1	2.1%	29	6.5%	*p* = 0.236
*Nontuberculosis mycobacteria*	1	2.7%	2	0.5%	*p* = 0.123
*S.maltophilia*	3	6.4%	26	5.6%	*p* = 0.820
*Achromobacter* spp.	1	2.1%	20	4.3%	*p* = 0.470
MRSA	2	4.3%	16	3.4%	*p* = 0.770
Respiratory function					
FEV_1,_ %	22	82.7 ± 21.1	252	85.2 ± 21.5	*p* = 0.638
FVC, %	23	80.8 ± 18.5	249	87.5 ± 19.9	*p* = 0.141
Nutritive status (Me (IQR); Me (Q25;Q75))					
BMI percentile		31.9 (58.7); 31.9 (9.3; 67.4)		32.8 (51.1); 32.8 (11.8; 62.7)	*p* = 0.918
Disease complications in the reporting year					
ABPA	1	2.1%	9	1.9%	*p* = 0.923
DIOS	1	2.2%	9	1.9%	*p* = 0.905
Pseudo-Bartter syndrome	3	6.4%	21	4.5%	*p* = 0.550
Pneumothorax	0	0.0%	3	0.6%	*p* = 0.581
Hemoptysis	0	0.0%	0	0.0%	-
*Occur malignancy*	0	0.0%	0	0.0%	-
Osteoporosis	0	0.0%	9	2.8%	*p* = 0.388
Upper respiratory tract polyposis	19	40.4%	170	37.0%	*p* = 0.639
Amyloidosis	0	0.0%	0	0.0%	-
Diabetes					
0—no diabetes	46	95.8%	463	97.5%	*p* = 0.709
1—yes, daily insulin treatment	2	4.2%	9	1.9%	
2—yes, treatment with tableted hypoglycemic medication	0	0.0%	1	0.2%	
3—yes, only diet	0	0.0%	2	0.4%	
Liver disease					
0—no liver disease	33	68.8%	381	80.2%	*p* = 0.055
1—cirrhosis with hypertension	4	8.3%	11	2.3%	
2—cirrhosis without hypertension	3	6.3%	15	2.3%	
4—liver disease without cirrhosis	8	16.7%	68	14.3%	
Therapy in the reporting year					
Inhalation with hypertonic saline	42	89.4%	390	83.2%	*p* = 0.272
Mannitol	3	6.0%	21	4.4%	*p* = 0.601
Antibiotic	23	48.9%	225	48.0%	*p* = 0.900
Antibiotic Intravenous	15	33.3%	137	29.4%	*p* = 0.581
Antibiotic oral	22	48.9%	261	55.8%	*p* = 0.375
Bronchodilators	22	46.8%	225	48.1%	*p* = 0.868
Oxygen therapy	1	2.0%	5	1.0%	*p* = 0.543
Dornase-alpha	49	100.0%	470	98.9%	*p* = 0.471
Steroid Inhaled	4	8.5%	54	11.5%	*p* = 0.531
Steroid Oral	0	0.0%	14	2.9%	*p* = 0.228
Azithromycin	9	20.0%	152	32.5%	*p* = 0.083
Ursodeoxycholic acid	48	100.0%	451	94.9%	*p* = 0.111
Pancreatic enzymes	48	100.0%	469	98.7%	*p* = 0.434
PPI	11	23.9%	103	22.0%	*p* = 0.767
Vitamin	43	93.5%	455	97.2%	*p* = 0.163
Kinesytherapy	43	91.5%	429	92.1%	*p* = 0.891
Fecal elastase					
<200 ng/g	30	100.0%	305	96.20%	*p* = 0.278
≥200 ng/g	0	0.0%	12	3.80%	
Therapy at home and in hospital					
Total IV therapy (at home + in hospital) in the reporting year	43	15.1 ± 17.2	420	14.0 ±23.2	*p* = 0.447
IV therapy in hospital in the reporting year	43	13.2 ± 16.0	420	11.9 ±16.3	*p* = 0.577
Total days in hospital in the reporting year	43	18.9 ± 18.3	432	16.0 ±18.6	*p* = 0.300

Note: MRSA, methicillin-resistant Staphylococcus aureus; BMI, body mass index; ABPA, allergic bronchopulmonary aspergillosis; DIOS, distal intestinal obstruction syndrome; PPI, proton pump inhibitors; IV therapy, intravenous therapy.

**Table 2 ijms-23-10377-t002:** Results of the intestinal current measurements method. Short-circuit current density (ΔISC) as a response to stimulators in a group of patients with cystic fibrosis, M ± m.

ΔISC, µA/cm^2^	Basal	Amiloride	Forskolin	Genistein	DIDS	Histamine
Healthy group (*n* = 18)	4.72 ± 6.3	−7.29 ± 9.84	26.43 ± 15.66	2 ± 0.29	1.8 ± 0.26	115.54 ± 59.49
Class I/class I (*n* = 3)	6.44 ± 1.24	−8.67 ± 2.09	0.25 ± 0.12	0.5 *	0.5 *	3.25 ± 0.31
F508del/F508del (*n* = 3)	10.06 ± 1.5	−18.39 ± 5.62	3.06 ± 0.89	1.83 ± 0.35	1.83 ± 0.35	21.5 ± 5.46
F508del/ [L467F;F508del] (*n* = 3)	7.44 ± 1.95	−11.39 ± 0.79	2.5 ± 0.61	0.5 *	0.5 *	12.39 ± 1.98

*—all patients showed the same result.

## Data Availability

The datasets used and/or analyzed during the current study are available from the corresponding author upon reasonable request.

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
