# Peer review of "Evaluation of the Complex p.[Leu467Phe;Phe508del] CFTR Allele in the Intestinal Organoids Model: Implications for Therapy"

_ijms, 2022, doi:10.3390/ijms231810377_

Round 1
Reviewer 1 Report
The manuscript by Kondratyeva et al. investigates the frequency and therapeutic relevance of the L467F/F508del complex allele in Russian patients with Cystic Fibrosis. The study points out that a relatively high percentage of Russian CF patients carry these mutations in cis at least in one allele on the F508del homozygote background. The authors provide evidence that the L467F mutation further diminishes the stability and function of the F508del CFTR protein by furnishing references and presenting their findings. They provide evidence that the here described complex allele minimally responds to the dual combination therapy (VX-770+VX-809 or VX-770+VX-661). The topic highlights individual differences in therapeutical responses toward highly effective modulators that should be considered in clinical settings. Generally, the experiments are well designed, and controls are in place.
Critics:
1. Please consider rephrasing the sentence in lines 58-61. “Complex alleles of the CFTR gene may affect the effectiveness of target therapy; the mechanisms may differ from increasing the pathogenicity in case of two simultaneous mutations at the same allele (e.g., CFTR proteolysis increase or rather protein folding impairment) to alterations in ligation or effectiveness of the CFTR modulator.”
The second part of this sentence is disconnected and hard to understand.
2. Please consider wording the sentence differently in lines 83-84 “FIS assay results allow to prescribe target therapy to patients with CF.” The outcome of the FIS assay may indicate effectiveness of highly effective modulator therapy and could provide guidance in prescribing target therapy to patients.
3. In line 85: please consider changing with forskolin to by forskolin
4. In line 86: please correct c-AMP to cAMP
5. In line 86 (the same line), please consider using: In intestinal organoids, spherical structures of…
6. In line 87: Water passively moves and does not transport. It follows sodium and chloride transport.
7. Along with the whole manuscript, please change the chlorine ion to chloride ion. Chlorine is the name of the atomic state of the material, not the name of the ionic form.
8. Please add the missing lines that show the treatment time in the Figures. (Figure 1 Carbachol and histamine; Figure 2 Forskolin/IBMX: Figure 3 Amiloride, Forskolin/IBMX, Genistein.)
9. Please consider rewriting the method of ICM Line 410-413: If air is present in the nozzles of the electrode tip, then the electrode will fail to detect any electric current. This is troubleshooting, and it is not calibration. Calibration is to compensate for the difference between electrodes and fluid resistance. Also, this instrument cannot be calibrated to vibration, accidental electrode contact, or other equipment working in the room. These all are sources of noise that can disturb recordings and introduce false positive amplitude change or cancel out signals. Recordings collected in such an experiment cannot be evaluated in many cases.
10. In line 422: Amiloride is an inhibitor of the epithelial sodium channel (ENaC) and is not a stimulator.
11. In line 423: Carbachol is responsible for a muscarinic receptor-linked Ca signal, not a calcium channel stimulant.
12. In line 425: Please rephrase this sentence. “The analysis was terminated after recording the basal short-circuit current.” The basal short-circuit current recording happens at the beginning of the experiment prior to any acute pharmacological manipulation. The tracing may return to the basal level, but the current usually remains at the forskolin-stimulated levels in the absence of a CFTR inhibitor. Please see the tracing in Figure 3.
Author Response
- Please consider rephrasing the sentence in lines 58-61. “Complex alleles of the CFTR gene may affect the effectiveness of target therapy; the mechanisms may differ from increasing the pathogenicity in case of two simultaneous mutations at the same allele (e.g., CFTR proteolysis increase or rather protein folding impairment) to alterations in ligation or effectiveness of the CFTR modulator.”
The second part of this sentence is disconnected and hard to understand.
Response 1: We have corrected the text: «Two simultaneous mutations at the same allele (сomplex allele) may reduce the effectiveness of target therapy by increasing pathogenicity of each other. As an example, additional cis-variant may increase CFTR proteolysis or increase folding disorders. Furthermore, presence of additional cis-variant may lead to the changes in structure of protein. These changes in protein's strcuture, in return, may lead to decrease or inability of protein's interaction with CFTR modulators resulting in reduced effectiveness of CTFR modulators as such [7]».
- Please consider wording the sentence differently in lines 83-84 “FIS assay results allow to prescribe target therapy to patients with CF.” The outcome of the FIS assay may indicate effectiveness of highly effective modulator therapy and could provide guidance in prescribing target therapy to patients.
Response 2: We have replaced it with the one you suggested (The results of the FIS assay may indicate effectiveness of highly effective modulator therapy and could provide guidance in prescribing target therapy to patients). Thank you!
- In line 85: please consider changing with forskolin to by forskolin
Response 3: corrected
- In line 86: please correct c-AMP to cAMP
Response 4: corrected
- In line 86 (the same line), please consider using: In intestinal organoids, spherical structures of…
Response 5: corrected
- In line 87: Water passively moves and does not transport. It follows sodium and chloride transport.
Response 7: We have corrected (In intestinal organoids, spherical structures of one-layer epithelium, the activation of the CFTR channel leads to swelling due to the transport of chloride ions into the internal cavity of the organoids (lumen).)
- Along with the whole manuscript, please change the chlorine ion to chloride ion. Chlorine is the name of the atomic state of the material, not the name of the ionic form.
Response 7: corrected. Thank you!
- Please add the missing lines that show the treatment time in the Figures. (Figure 1 Carbachol and histamine; Figure 2 Forskolin/IBMX: Figure 3 Amiloride, Forskolin/IBMX, Genistein.)
Response 8: We have added the missing lines.
- Please consider rewriting the method of ICM Line 410-413: If air is present in the nozzles of the electrode tip, then the electrode will fail to detect any electric current. This is troubleshooting, and it is not calibration. Calibration is to compensate for the difference between electrodes and fluid resistance. Also, this instrument cannot be calibrated to vibration, accidental electrode contact, or other equipment working in the room. These all are sources of noise that can disturb recordings and introduce false positive amplitude change or cancel out signals. Recordings collected in such an experiment cannot be evaluated in many cases.
Response 9: Corrected: At the first stage, each recirculation chamber is calibrated separately using the VCC MC 8B421 Physiologic Instrument, San Diego, USA. Problems such as air in agar contact tips and fluid resistance were eliminated during chamber preparation. Noise sources: vibrations near the equipment, accidental contacts with electrodes, foreign working devices in the office were eliminated as far as possible.
- In line 422: Amiloride is an inhibitor of the epithelial sodium channel (ENaC) and is not a stimulator.
Response 10: Corrected in the text.
- In line 423: Carbachol is responsible for a muscarinic receptor-linked Ca signal, not a calcium channel stimulant.
Response 11: Corrected in the text.
- In line 425: Please rephrase this sentence. “The analysis was terminated after recording the basal short-circuit current.” The basal short-circuit current recording happens at the beginning of the experiment prior to any acute pharmacological manipulation. The tracing may return to the basal level, but the current usually remains at the forskolin-stimulated levels in the absence of a CFTR inhibitor. Please see the tracing in Figure 3.
Response 12: Corrected: the analysis was terminated after recording the final resistance of tissue (Rt, ohm.cm2).
Reviewer 2 Report
Dear Elena Kondratyeva et al.,
thank you for the opportunity to read your manuscript about the efficacy of individual treatment options with CFTR modulators for F508del/F508del-CF patients with the compound allele [Leu467Phe;Phe508del]. The information you gathered is very important and should, of course, be published.
I do, however, have some major problems in understanding the manuscript and what you really did and measured. Therefore, please consider my comments below to improve the text. thank you!
Major comments:
lines 112-115: table 1, liver disease, p-values: the only p-value you give is for the data about no liver disease (p=0,055). p=0,055 is NOT below 0,05! so there is really no statistically significant difference in table 1 whatsoever. (Similar issue lines 256-259)
lines 121-161: data is missing. In line 421 you describe the measurement of the pre-amiloride currents. This would be very informative to know, as without a functioning CFTR the basal current should already be different in the patient groups. Please include this data in your table 2 and the respective figures. (There might be a misunderstanding about the ICM and ΔISC-issue – see below.)
table 2, 3rd row: nowhere in your text you explained about this patient group (Class I/class I). Where do they come from? Which mutations in the CFTR gene do they have? Why do they have the same reaction to amiloride as the healthy controls?
table 2 and figures 1-3: What is the difference between ICM and ΔISC? I did not find the explanation in the materials and methods section. If ΔISC is ICM with stimulator – ICM at baseline (pre-amiloride condition) this needs to be specified somewhere.
lines 131-133, lines 142-143, lines 155-157, figures 1-3: I do not understand your comment. As the data here stems from only 3 different persons each – and the data in the figures come from one person only – how can you not control for amiloride-intake? Is the measurement in the graphs shown for a person who takes amiloride or not? Why mention it, if not? Did any of the three in the table take amiloride? Doesn’t that completely corrupt the data?
Please explain, if the figure starts with background (pre-amiloride condition) and the lines present the times, when stimulators are first introduced into the system.
to both of the above: This makes no sense, as in figure 1 there is hardly any reaction to amiloride, while in table 2 the strongest reaction to amiloride is seen exactly in that group of patients???
lines 144-147: concerns same as above: if ΔISC (table 2 data) show the difference to baseline, then the response of the complex allele group is higher than in healthy controls and class I patients, but lower than in F508del homozygous persons.
table 2 and figures 1-3: why show the data of genistein (unspecific PAIN, not at all specific modulator of the CFTR), carbachol (a cholinomimetic drug that binds and activates acetylcholine receptors), DIDS (no idea what that means) and histamine (activates histamine receptors)? These data are not explained / discussed in your manuscript at all – although they show the most differences between healthy controls and CF patients. Why? The differences in ENaC function and in the response to forskolin should be enough to describe CFTR function.
lines 159-161: If I am not mistaken, class I mutations in the CFTR describe mutations that lead to complete absence of CFTR protein synthesis. If you say, your data show absence of CFTR function in the complex allele group – why is the data then so different from the class I data? Please concentrate your table, figures and text on the effect of forskolin – which is the only one really consistent with your discussion.
lines 256-259: table 1, liver disease, p-values: the only p-value you give is for the data about no liver disease (p=0,055). No significant difference was, therefore, detected.
lines 331-447: the headings “materials” and “methods” seem to be arbitrarily chosen here. People and their tissues are no “material” and for most chemicals (which only appear in the methods section) you have not specified the supplier. Lines 442-447 should appear under the sub header “statistics”.
lines 332-336: Please explain how you chose the 72.5% of the homozygous F508del/F508del-patients from the Russian CF register and why the other 17.5% were excluded.
line 375 and lines 403-427: you did NOT measure intestinal potential differences but currents.
Minor comments:
In general, please shorten the theory by focussing on the problem of complex alleles and [Leu467Phe;Phe508del] specifically in CF and sort introduction and discussion. The introduction should include CF, complex alleles, target therapy. When explaining the modulators in your introduction, please explain the meaning of the terms “corrector” and “potentiator”, specify which drug is which and why for F508del both are needed. The discussion, on the other hand, should focus on the validity of the assays used and your results in the context of results in the literature.
line 40: ...genetic variants. (s is missing)
line 61: for me “ligation” is the covalent binding of a molecule to the channel. Maybe you meant “binding”?
lines 80-90: should go to the discussion
table 1: all used abbreviations need to be explained, and not just somewhere in the following text, but in the subscript of the table
table 1, age of diagnosis, M+/-SD: can you really have 0.6 years minus 1 year? Is the raw data really normally distributed?
table 2: I, personally, would find it much easier to read, if the sequence of the rows were inverted, starting with the healthy controls, then class-I-patients, F508del and last the complex allele patient data.
figures 1-3: same sequence of the figures as in the table– or even better: 1 big figure with parts a-d;
I am missing the figure for the data of the class-I-patients.
Why do you use similar x-axis spreads in your figures 1-3? It should be either exactly the same, as to be able to better compare the three, or you should use the optimal spread on each single plot. It is now very difficult to appreciate optically the differences between figure 1 and 2.
figures 1-3 and line 424: what is DIDS? (Abbreviation not spelled out)
line 170 and figure 4: If only data of the 5µM concentration is discussed, why use 0.128µM forskolin, too?
line 200: why not mention culture 2?
lines 218-245: this is introduction – not discussion of your results; and all the other complex alleles are not the topic of this manuscript and specifics about them should be omitted.
lines 271-274: only the forskolin data show the discussed effect – I do not see the same effect on ENaC (sodium channel).
line 281: last word in line should be a “the” not an “a”
line 282-283: tissue cannot be collected from living people in vitro
line 343: just any patients or the exact patients, that were used for table 1 data?
line 481: delete “. at the end
Author Response
Response to Reviewer 2 Comments
Major comments:
- lines 112-115: table 1, liver disease, p-values: the only p-value you give is for the data about no liver disease (p=0,055). p=0,055 is NOT below 0,05! so there is really no statistically significant difference in table 1 whatsoever. (Similar issue lines 256-259)
Response 1: We agree with your comment and have made changes.
- lines 121-161: data is missing. In line 421 you describe the measurement of the pre-amiloride currents. This would be very informative to know, as without a functioning CFTR the basal current should already be different in the patient groups. Please include this data in your table 2 and the respective figures. (There might be a misunderstanding about the ICM and ΔISC-issue – see below.)
Response 2: added
- table 2, 3rdrow: nowhere in your text you explained about this patient group (Class I/class I). Where do they come from? Which mutations in the CFTR gene do they have? Why do they have the same reaction to amiloride as the healthy controls?
Response 3: the class 1/class 1 group consisted of patients with class 1 variants in the genotype: 2143delT/712-1G->T, G542X/R785X, c.264_268del/ 3271+1G>T. Added in the text.
- table 2 and figures 1-3: What is the difference between ICM and ΔISC? I did not find the explanation in the materials and methods section. If ΔISC is ICM with stimulator – ICM at baseline (pre-amiloride condition) this needs to be specified somewhere.
Response 4: ICM is the name of the test method and ΔIsc is the short circuit current. Basal pre-amiloride is also included in the table 2.
- lines 131-133, lines 142-143, lines 155-157, figures 1-3: I do not understand your comment. As the data here stems from only 3 different persons each – and the data in the figures come from one person only – how can you not control for amiloride-intake? Is the measurement in the graphs shown for a person who takes amiloride or not? Why mention it, if not? Did any of the three in the table take amiloride? Doesn’t that completely corrupt the data?
Please explain, if the figure starts with background (pre-amiloride condition) and the lines present the times, when stimulators are first introduced into the system.
to both of the above: This makes no sense, as in figure 1 there is hardly any reaction to amiloride, while in table 2 the strongest reaction to amiloride is seen exactly in that group of patients???
Response 5: There is a misprint in the text. The addition of amiloride proceeds according to the SOPs. Patients do not receive amiloride as a drug.
It was: Intestinal current measurement (ICM); patient with a F508del/[L467F;F508del] genotype. Comment – in case of amiloride intake ΔISC decreased, no response to forskolin/IBMX, and as a response to carbachol and histamine ΔISC changed to negative.
It became: Intestinal current measurement (ICM); patient with a F508del/F508del genotype. Comment – the addition of amiloride caused a decrease in the short-circuit current density (ΔISC) was detected, no response to forskolin/IBMX, and as a response to histamine ΔISC changed to negative.
The table shows the average data from several patients of patients (M±m), and the figure shows the result from the 1st randomly selected patient. The mean is high and the figure is not a very high response, also there is interference in the figure before adding forskolin, so the response does not seem to be that high.
- lines 144-147: concerns same as above: if ΔISC (table 2 data) show the difference to baseline, then the response of the complex allele group is higher than in healthy controls and class I patients, but lower than in F508del homozygous persons.
Response 6: Changed in the text.
- table 2 and figures 1-3: why show the data of genistein (unspecific PAIN, not at all specific modulator of the CFTR), carbachol (a cholinomimetic drug that binds and activates acetylcholine receptors), DIDS (no idea what that means) and histamine (activates histamine receptors)? These data are not explained / discussed in your manuscript at all – although they show the most differences between healthy controls and CF patients. Why? The differences in ENaC function and in the response to forskolin should be enough to describe CFTR function.
Response 7: DIDS - 4,4'-diisothiocyano-2,2'-stilbene-disulfonic acid.
- lines 159-161: If I am not mistaken, class I mutations in the CFTR describe mutations that lead to complete absence of CFTR protein synthesis. If you say, your data show absence of CFTR function in the complex allele group – why is the data then so different from the class I data? Please concentrate your table, figures and text on the effect of forskolin – which is the only one really consistent with your discussion.
Response 8: The complex allele L467F;F508del is located on only one allele, the second allele has the F508del mutation, so the results differ from genotypes with class I mutations only.
- lines 256-259: table 1, liver disease, p-values: the only p-value you give is for the data about no liver disease (p=0,055). No significant difference was, therefore, detected.
Response 9: We agree with your comment and have made changes
- lines 331-447: the headings “materials” and “methods” seem to be arbitrarily chosen here. People and their tissues are no “material” and for most chemicals (which only appear in the methods section) you have not specified the supplier. Lines 442-447 should appear under the sub header “statistics”.
Response 10: We agree with your comment and have made changes
- lines 332-336: Please explain how you chose the 72.5% of the homozygous F508del/F508del-patients from the Russian CF register and why the other 17.5% were excluded.
Response 11: This group consisted of children under 2 years of age and patients of other age groups who did not sign an informed consent for the study. Coverage is sufficient to draw conclusions
- line 375 and lines 403-427: you did NOT measure intestinal potential differences but currents.
Response 12: A misprint. Changed to intestinal current measurement. Thank you
Minor comments:
- In general, please shorten the theory by focussing on the problem of complex alleles and [Leu467Phe;Phe508del] specifically in CF and sort introduction and discussion. The introduction should include CF, complex alleles, target therapy. When explaining the modulators in your introduction, please explain the meaning of the terms “corrector” and “potentiator”, specify which drug is which and why for F508del both are needed. The discussion, on the other hand, should focus on the validity of the assays used and your results in the context of results in the literature.
Response 13. Thank you for your valuable comments, but we would like to clarify that the impact of any complex alleles on the effectiveness of targeted therapy in cystic fibrosis is just beginning to be explored and we have focused on this in the introduction and discussion, and data on the allele is not numerous [Leu467Phe; Phe508del].
- line 40: ...genetic variants. (s is missing)
Response 14: Corrected
- line 61: for me “ligation” is the covalent binding of a molecule to the channel. Maybe you meant “binding”?
Response 15: Completely rewritten proposal “These changes in protein's structure, in return, may lead to decrease or inability of protein's interaction with CFTR modulators resulting in reduced effectiveness of CTFR modulators as such”
- lines 80-90: should go to the discussion
Response 16: Corrected
- table 1: all used abbreviations need to be explained, and not just somewhere in the following text, but in the subscript of the table.
Response 17: we agree with your comment and have made changes
- table 1, age of diagnosis, M+/-SD: can you really have 0.6 years minus 1 year? Is the raw data really normally distributed?
Response 18: Corrected
- table 2: I, personally, would find it much easier to read, if the sequence of the rows were inverted, starting with the healthy controls, then class-I-patients, F508del and last the complex allele patient data.
Response 19: changed
- figures 1-3: same sequence of the figures as in the table– or even better: 1 big figure with parts a-d;
I am missing the figure for the data of the class-I-patients.
Why do you use similar x-axis spreads in your figures 1-3? It should be either exactly the same, as to be able to better compare the three, or you should use the optimal spread on each single plot. It is now very difficult to appreciate optically the differences between figure 1 and 2.
Response 20: added and changed
- figures 1-3 and line 424: what is DIDS? (Abbreviation not spelled out)
Response 21: DIDS - 4,4'-diisothiocyano-2,2'-stilbene-disulfonic acid. Also added to the text.
- line 170 and figure 4: If only data of the 5µM concentration is discussed, why use 0.128µM forskolin, too?
Response 22: We have added the text: «When F508del/[L467F;F508del] organoids were treated by forskolin at 0.128 μM concentration the effects of VX-770, VX-809, VX-770+VX-809, VX-770+VX-661 on the rescue of CFTR function were insignificant and the AUC values did not exceed 1000 (Figure 4). Only the combination of VX-770+VX-661+VX-445 resulted in the rescue of CFTR function (Figure 4, organoid culture 2 and 3)».
- line 200: why not mention culture 2?
Response 23: We have added information on culture 2.
- lines 218-245: this is introduction – not discussion of your results; and all the other complex alleles are not the topic of this manuscript and specifics about them should be omitted.
Response 24: Thanks for the comment, but the data on the Leu467Phe allele; Phe508del] are not very numerous, and in the discussion we wanted to draw attention to the problem of the effectiveness of targeted therapy, taking into account the possible negative impact of complex alleles.
- lines 271-274: only the forskolin data show the discussed effect – I do not see the same effect on ENaC (sodium channel).
Response 25: corrected
- line 281: last word in line should be a “the” not an “a”
Response 26: corrected.
- line 282-283: tissue cannot be collected from living people in vitro
Response 27: corrected.
- line 343: just any patients or the exact patients, that were used for table 1 data?
Response 28: corrected.
- line 481: delete “. at the end
Response 29: corrected. Thank you very much for reading and carefully analyzing the manuscript of the article!